# Pharmacologic Management for Ventricular Arrhythmias: Overview of Anti-Arrhythmic Drugs

**DOI:** 10.3390/jcm11113233

**Published:** 2022-06-06

**Authors:** John Larson, Lucas Rich, Amrish Deshmukh, Erin C. Judge, Jackson J. Liang

**Affiliations:** 1Division of Internal Medicine, University of Michigan, Ann Arbor, MI 48109, USA; lajohn@med.umich.edu (J.L.); lurich@med.umcih.edu (L.R.); 2Division of Cardiovascular Medicine, Cardiac Arrhythmia Service, University of Michigan, Ann Arbor, MI 48109, USA; damrish@med.umich.edu (A.D.); rerin@med.umich.edu (E.C.J.)

**Keywords:** ventricular arrhythmias, ventricular tachycardia, ventricular fibrillation, anti-arrhythmic drugs, electrical storm

## Abstract

Ventricular arrhythmias (Vas) are a life-threatening condition and preventable cause of sudden cardiac death (SCD). With the increased utilization of implantable cardiac defibrillators (ICD), the focus of VA management has shifted toward reduction of morbidity from VAs and ICD therapies. Anti-arrhythmic drugs (AADs) can be an important adjunct therapy in the treatment of recurrent VAs. In the treatment of VAs secondary to structural heart disease, amiodarone remains the most well studied and current guideline-directed pharmacologic therapy. Beta blockers also serve as an important adjunct and are a largely underutilized medication with strong evidentiary support. In patients with defined syndromes in structurally normal hearts, AADs can offer tailored therapies in prevention of SCD and improvement in quality of life. Further clinical trials are warranted to investigate the role of newer therapeutic options and for the direct comparison of established AADs.

## 1. Introduction

Ventricular arrhythmias (VAs) including ventricular tachycardia (VT) and ventricular fibrillation (VF) can be life-threatening and can adversely affect patient quality of life [1,2]. The gold standard for prevention of sudden cardiac death (SCD) is implantable cardiac defibrillator (ICD) implantation. While ICDs remain a cornerstone of VA therapy, both appropriate and inappropriate shocks in patients are associated with higher morbidity and mortality with a corresponding lower quality of life [3,4,5]. In patients treated for secondary prevention of VA, recurrent VA leading to device therapies occur in approximately 20–50% of patients within 5 years [6]. While ICD therapies abort sudden death and prolong survival, palliation of recurrent VA is accomplished through antiarrhythmic drugs (AADs) and catheter ablation. AADs are an important treatment option for long-term management of recurrent VA [7,8]. This review focuses on AAD management of recurrent VA. We also discuss the use of AADs in life threatening situations such as electrical storm (ES).

The majority of patient data and discussion will focus on VAs in patients with abnormal heart structure, but this review also includes discussions on defined syndromes of sustained VAs including Brugada Syndrome, Long QT Syndrome, and Catecholaminergic Polymorphic VT.

## 2. Drug Classification

The Vaughan–Williams (VW) classification system is the most commonly used system to classify AADs. It has the advantages that AADs are clustered by channel, effect on the action potential, and effect on parameters within an EP study (sinus node function, AV conduction, QT). Figure 1 illustrates the effect each medication class has on the cardiac action potential and corresponding EKG tracing. Also included are medications that do not fit within the classification of this system. As a result, the clinical utility and hazards of similarly classified medications often align [9]. Despite this, the VW system has limitations. While broadly organized by mechanism of action, further study of anti-arrhythmic pharmacology has shown a wide variation of mechanisms within classes. Importantly, three notable antiarrhythmic medications (ranolazine, ivabradine, and digoxin) do not fall within this classification system. Further modernized versions of the VW classification system have been proposed to incorporate these three medications [10]. Table 1 below details the class, mechanism of action, channel target, and major side effects/contraindication of each medication.

## 3. Class I Medications: Sodium Channel Blockade

All of the Class I Medications affect the cardiac electrical system via the blockade of sodium channels. Despite this, the blockade of different sodium channels can have a wide variety of effects, and therefore, this overall class is split into three subgroups: Class IA, Class IB, and Class IC. All Class I medications bind in a “use-dependent” fashion. This use-dependent behavior causes increased drug binding and consequently increased sodium channel blockade at faster stimulation rates (i.e., faster heart rates) [11]. Pharmacologically, this is advantageous in the management of Vas, as these medications become more potent at higher heart rates.

### 3.1. Class IA

The Class 1A AADs exhibit similar electrophysiologic effects via blockade of the rapid inward sodium depolarization current in a use-dependent fashion and prolongation of repolarization via blockade of the delayed rectifier potassium channel in a reverse use-dependence fashion [12,13]. Furthermore, the slow phase 4 depolarization occurs during spontaneous automaticity [14]. The net effect is preferential prolongation of action potential duration at fast heart rates, prolonged effective refractory period (ERP), and decreased automaticity.

#### 3.1.1. Procainamide

There have been multiple studies showing the efficacy of procainamide as an adjunct medication in patients with ongoing Vas, despite more commonly used AADs (Amiodarone or Lidocaine) in acute VA [15,16,17]. The PROCAMIO trial randomized patients in stable VT to either intravenous procainamide or to intravenous amiodarone. Termination of VT occurred in 67% of patients treated with procainamide versus 38% of amiodarone patients. VT termination was defined as return to previous underlying rhythm within 40 min of drug administration. Importantly, there were also fewer adverse events at both 40 min and 24 h in the procainamide group (9% vs. 41% in 40 min, 18% vs. 31% in 24 h) [17]. A trial by Gorgels et al. showed a similar result for procainamide vs. lidocaine. In a group of 30 patients, 21% of lidocaine-treated patients had termination of VT as compared to 80% of procainamide treated patients [16]. These trials indicate that procainamide is a safe and effective option for management of VA and ES.

In contrast, the use of oral procainamide for outpatient prevention of secondary VA has limited evidentiary support. While the 1993 ESVEM trial data largely supported the use of Sotalol, it also found the procainamide was an effective agent at controlling VAs and limiting the inducibility of VAs in subsequent EP studies [18]. Similar to its use in acute situations, procainamide has largely been studied as an AAD in patients with VT refractory to amiodarone. In this setting, a 34-patient study by Toniolo et al. showed that use of procainamide after failure of previous AAD resulted in 10% reduction of ICD therapies [19]. Of note, this study has a high proportion of concomitant ablations (18 of 34 patients) that may serve as a significant confounder. In two older single center studies, procainamide showed efficacy in the prevention of recurrent arrhythmias in small cohorts [20,21]. While these small studies show relatively few episodes of discontinuation, procainamide also acts as a negative inotrope and has been shown to cause a significant amount of hypotension, especially in its parenteral form [17,21]. Procainamide has also gained popularity as a provocative medication to diagnose Brugada syndrome [22].

#### 3.1.2. Quinidine

Given the efficacy of procainamide, quinidine, which is more readily available in oral formulations, has been used as a salvage therapy for patients with structural heart disease for recurrent VA despite prior AAD treatment. It has also shown to have efficacy in both idiopathic and post-infarct ectopy-triggered polymorphic VT (PMVT) syndromes [23,24]. A study by Viskin et al. showed that 43 patients with acute MI and multi-drug resistant PMVT had a reduction in mortality and lower rates of recurrent VT after quinidine treatment [24]. This observation remained true in a small 20 patient observation study in patients with recent discharge after revascularization and PVMT [25]. On the contrary, the efficacy of quinidine as salvage therapy for recurrent monomorphic VT in patients with structural heart disease who have failed other AADs is limited, and frequent adverse effects may lead to discontinuation [26,27,28].

Quinidine’s use as a primary agent in VA that may have a component of ectopy-triggered sustained arrhythmias, namely Brugada Syndrome, has been well studied. While the underlying pathophysiology of Brugada remains poorly understood, ventricular ectopy can lead to sustained and life-threatening VA in these patients [29]. In multiple small studies, quinidine has been consistently shown to reduce ICD shocks and improve morbidity (although mortality benefit in ICD populations is less clear) [29,30,31,32]. A study by Belhassen et al. showed that in 25 patients with known Brugada Syndrome (15 symptomatic and 10 with inducible VT by EP study), 88% of patients treated with quinidine had no further VAs. Furthermore, in 19 patients who were still on quinidine after 6 months, none had any further VAs over an average follow-up time of 56 months [32]. Given the low prevalence of Brugada syndrome, these studies reflect small patient populations and are a collection of prospective and retrospective cohort studies.

#### 3.1.3. Disopyramide

With further data on procainamide and quinidine, the use of disopyramide has largely fallen out of clinical practice as an AAD. Its known effect of negative inotropy and proarrhythmic qualities at high dosages have been strong factors discouraging its use. However, older studies utilizing this medication are still informative for clinical decision-making in areas where disopyramide is readily available. Its disuse has also been driven by studies showing decreased efficacy when compared to alternative AADs [20,33,34,35,36]. Within disopyramide arms of early multi-AAD studies and in disopyramide-alone studies, disopyramide showed improvement in Vas with minimal side effects [35,36,37]. Notably, these early studies used reduction in PVC burden as primary endpoints, and reduction in sustained VT, ICD shocks, and general mortality was not assessed.

Due to its negative inotropic effects, disopyramide is frequently used in patients with hypertrophic cardiomyopathy (HCM) and LV outflow tract obstruction. Some observational evidence may suggest a decreased mortality secondary to SCD or VAs in patients appropriately considered for disopyramide therapy [38]. However, there have been no studies on this population and prevention of SCD or secondary prevention of Vas, and in some studies > 40% of HCM patients self-discontinued the medication due to side effects [39].

### 3.2. Class IB

The Class IB AADs, lidocaine and mexiletine, exhibit a use-dependent blockade of the inward sodium depolarization current, thereby decreasing maximal velocity of depolarization, shortening of action potential and ERP duration, and decreasing automaticity of phase 4 depolarization [40].

#### 3.2.1. Lidocaine

Lidocaine is only available in intravenous form and remains one of the most heavily used agents in the acute inpatient treatment of VAs [41]. Despite its frequent usage, there remains little evidence to support its efficacy. Lidocaine was initially used prophylactically for prevention of VAs after acute MI, but a meta-analysis by Hine et al. found that pre-hospital administration showed no benefit while in-hospital prophylaxis had increased mortality with no significant reduction in VAs [42]. In trials comparing it to either procainamide or amiodarone, lidocaine was shown to be inferior to both medications [17,43]. The 2016 ROC-ALPS trial showed that in patients with out-of-hospital arrests due to refractory VF or VT arrest, amiodarone showed the best efficacy, but neither amiodarone of lidocaine had a statistical improvement in survival compared to placebo [44].

#### 3.2.2. Mexiletine

The foundation for the frequent use for mexiletine therapy was established through efficacy demonstrated in multiple small studies in the 1980s and early 1990s [45,46,47,48,49,50]. Later evidence showed that mexiletine was less efficacious in comparison to amiodarone as first line AAD for VA. This was solidified in a 2015 Cochrane review discussed more thoroughly in the amiodarone section [51]. Despite this, mexiletine continues to have a role in therapy as both an alternative in patients with toxicity to Class III agents and as combination therapy in refractory patients. In two studies of patients with refractory VT and amiodarone use, the use of mexiletine showed a significant reduction in sustained VT/VF and ICD therapies [52,53]. Both represented cohort studies (with sizes of 17 and 34 patients respectively) in which catheter ablation was either ineffective or contraindicated. In one study by Gao et al., 29 patients with refractory VT on amiodarone therapy underwent the addition of mexiletine. In comparison to matched observational time prior to initiation, there was a reduction of sustained VT/VF (median 12 vs. 2, *p* = 0.001) and ICD therapies (median 0 vs. 2, *p* = 0.003) [54]. Again, these represent observational trials with no control group or randomization. Despite mexiletine’s frequency of use as both an alternative and adjunctive agent, there is a profound lack of data supporting these strategies. The VANISH trial by Sapp et al. showed that escalation of anti-arrhythmic (largely mexiletine) was beneficial in only 32% of patients and was clinically inferior to alternative strategies such as catheter ablation [55].

An area with relatively strong support for mexiletine is in patients with Congenital Long QT Syndrome Type 3 (LQT3). The genetic gain-of-function mutation effects sodium channel NaV1.5. The result is a prolongation of QT through delayed sodium channel current which Mexiletine effectively blocks. The largest trial by Mazzanti et al. showed that 34 LQT3 patients treated with Mexiletine saw a reduction in not only QT intervals, but also an 87% relative reduction in patients with VAs (22 to 3%) and a 93% reduction in arrhythmia event rate [56].

### 3.3. Class IC

In regard to the effect on their target channel, Class IC AAD are the most potent among the inward sodium blocking agents. The strength of this binding thereby markedly reduces the action potential conduction velocity in atrial, ventricular, and specialized conduction tissues in a use-dependent fashion with minimal effect on overall action potential duration or ERP [57].

#### Flecainide and Propafenone

Given the limited efficacy and use of these medications in chronic VAs, especially in the setting of the relative contra-indications in patients with structural heart disease, we will discuss both flecainide and propafenone in this section. Both the CAST trial and the CASH trial showed a significantly elevated mortality in patients with structural heart disease treated with these agents [58,59]. The CAST trial showed a nearly 400% increase in arrhythmic death in patients treated with Class IC agents. Because of this, these agents have been typically reserved for used in the treatment of VAs in patients with structurally normal hearts. Catecholaminergic polymorphic ventricular tachycardia (CPVT) is a rare channelopathy characterized by exercise or adrenergic-induced polymorphic VT frequently manifesting in childhood. While beta blockers are the mainstay of therapy in CPVT given a larger evidence base, flecainide and propafenone have been shown to be effective as a secondary medication in refractory patients [60,61]. In the largest observational trial by Roston et al., 25% of patients treated with beta blockers experienced treatment failure. Subsequently, 36 patients with beta blocker treatment failure went on to receive flecainide therapy, and of those, only 6% (3 patients) went on to have further failure [60]. Further randomized control trials in this area are ongoing. Comparably, Class IC agents have been explored as a secondary option in the treatment of Long QT Syndrome, especially in LQT3 [62,63]. While there is limited evidence, a small, randomized trial demonstrated efficacy in both reduction of events and decrease in QT interval when compared to placebo [63].

Lastly, similar to procainamide, while flecainide should not be used to prevent VA in patients with Brugada syndrome, it has been shown to have a useful role in helping to establish a diagnosis [23].

## 4. Class II Medications: Beta Blockade

Beta blockers exhibit their antiarrhythmic properties via blunting sympathetic activity on cardiac tissue, most notably through decreasing phase 4 depolarization and thereby decreasing automaticity and conduction velocity [64]. Beta blocker medications have long served as a cornerstone of congestive heart failure therapy. A component of this use has been the reduction in mortality secondary to SCD [1,65,66,67,68]. A proposed mechanism of the decreased SCD risk is via the reduction of both primary and secondary VAs. This effect has been well established in patients with hospitalizations for ACS. In one of the major initial trials for beta blockers, the BHAT trial, there was reduction in both cardiac death and VA in patients treated with propranolol after an acute MI [69]. In the CAPRICORN trial, patients who had the addition of carvedilol after an acute myocardial infarction had a 4.3 times relative risk reduction of VAs including a 70% reduction in VT [68]. Since this initial evidence, further trials have supported the use of beta blockers in recurrent VAs. Initial studies prior to the development of ICDs showed that in a small cohort, high doses of propranolol were effective at preventing symptomatic, recurrent VT [70]. In a similar study in 1989, 30 patients who underwent treatment with beta blocker alone had a significant reduction in inducible VT, VT on long-term monitoring, and VT during exercise [71]. Of 24 patients on long-term monotherapy, only six had recurrent VT after mean follow-up of 1068 days. In a later study by Levine et al., 218 patients who underwent ICD implantation were followed to evaluate for modifiable factors for initial and recurrent discharges. Beta blocker administration was found to significantly prolong time to first shock as well as significantly reduce the number of secondary shocks [72]. Despite these early promising results, large ICD trials have shown that with the introduction of life-saving ICD therapy, beta blocker therapy does not have an impact on mortality [66,68].

Conversely, studies on the impact of beta blocker therapy on the morbidity of VAs and ICD therapies show improvement. The OPTIC trial showed that patients on beta blocker therapy with ICD devices had a reduction in ICD therapies [73]. Even more importantly, the OPTIC study supported beta blockers role as an adjunct therapy. In comparison to beta blockers alone, beta-blocker plus amiodarone had a 12% absolute risk reduction and 37% relative risk reduction in ICD shocks. This combination effect has been reproduced in other trials. In a post hoc analysis of two pooled clinical trials, EMIAT and CAMIAT trials, the combination of beta-blocker and amiodarone led to a significant reduction in secondary arrhythmic death [74]. Given these benefits, beta blockers are now recommended as adjunct therapy in major guidelines for treatment of secondary prevention of VAs in patients with ischemic cardiomyopathy (ICM) [1].

This improvement in combination is also seen with class I agents. Patients in the CAST trial showed a VA reduction in patients treated with both beta-blockers and class IC agents [75]; this improvement has also been seen in other small, earlier trials [76,77]. The underlying mechanism has been hypothesized to be the reduction of proarrhythmic side effects in some patients on class 1C medications. This “anti-proarrhythmic” effect may allow for a more tailored use of alternative agents in specific patients [78].

Despite its limitations in the general treatment of recurrent arrhythmias, there is clear evidence for the use of beta blockers in specialized circumstances. In acute ES, beta blocker therapy remains a cornerstone of dampening adrenergic stimulation [79]. Multiple trials in ES have shown continued efficacy in the reduction of ICU days, reduction of shocks, and reduction of overall mortality [80,81]. Furthermore, the use of non-selective beta blockers, specifically propranolol, has been shown to have superior outcomes compared to selective beta blockers [81,82,83]. While there was no significant reduction in mortality, there was reduction in VA incidence as well as ICU admission time in patients treated with propranolol compared to metoprolol. In CPVT, beta blocker therapy is an evidence-based cornerstone of therapy [84,85,86]. Current guidelines recommend nadolol as the preferred long-acting beta blocker for VT prevention [86]. While the use of non-selective beta blockers should be considered in all patients with history of VA, the data only directly support this use in ES and CPVT.

Lastly, beta blockers have been shown to have a role in Long QT syndrome as well. The strongest evidence has been seen in types LQT1/LQT2, as beta blockers have been seen to historically reduce syncopal events and SCD while more modernly showing shorten QT intervals at faster rates [87,88]. However, beta blockers remain the first line therapy for treatment of LQT3 as well with less support [1]. A smaller observation trial by Wilde et al. showed that the use of beta blockers significant reduced arrhythmic events, although this trial was notable for the low number of arrhythmic events [89].

## 5. Class III Medications: Potassium Channel Blockade

### 5.1. Amiodarone

Amiodarone functions primarily through blockade of the delayed rectifier potassium channel, effectively prolonging repolarization and increasing the ERP to decrease the likelihood of reentry [90]. Unlike other agents in class III as discussed below, amiodarone does not exhibit reverse use dependence and therefore does not have similar rates of bradycardia and increased efficacy at lower heart rates [90]. While amiodarone has been shown to prolong QT interval, long-term safety studies do not show a corresponding significant increase in torsades de pointe [91,92]. This safety profile persisted in patients with a history of torsades to pointe [91,92]. Additionally, amiodarone has mechanistic overlap with all other classes of antiarrhythmics and has vasodilatory and negative inotropic effects.

Amiodarone is the most studied and most commonly used medication in the treatment of secondary VAs. It remains the only guideline-directed medication in patients with either ischemic or non-ischemic cardiomyopathy [1]. Prior to the popularization of ICDs and the AVID trial, trials such as the EMIAT trial and CAMIAT trial strongly supported amiodarone as a primary prevention strategy in ischemic heart disease [5,93,94]. With ICDs now the backbone of primary prevention strategy, prophylactic amiodarone use is an uncommon practice. However, in the setting of secondary prevention, amiodarone has been a long-established therapy. Early studies such as CASH and CASCADE study showed a reduction in both recurrent VAs and arrhythmic death; these findings were supported by a 1997 amiodarone trials meta-analysis [58,93,95]. More recently, the 2011 ALPHEE study showed that in prevention of ICD shocks, amiodarone had a statistically significant 16% absolute risk and 26% relative risk reduction as compared to placebo [96]. When combined with beta blockers, amiodarone has been shown to have increased efficacy. The previously mentioned OPTIC and pooled EMIAT and CAMIAT analysis showed that the addition of amiodarone to beta blocker alone had a 37% relative risk reduction and statistically significant reduction in ICD shocks [73,74]. In its parenteral form, amiodarone has been shown to decrease inpatient ventricular arrhythmias by up to 40% [97]. In ES, amiodarone has been shown to have superior efficacy to lidocaine [44]. In a comparative trial by Greene et al., the use of amiodarone has also been shown to reduce the recurrence of ES by nearly 50% over 5 years [98]. While these former trials showed a reduction in ES or VT/VF, the more recent ROC-ALPS study showed that in patients with recurrent out-of-hospital VT/VF arrest, both amiodarone and lidocaine did not show superiority to placebo in survival-to-admission rates [45]. Similarly, while amiodarone has been shown to reduce ICD shocks and recurrent VA, a 2015 Cochrane review showed that amiodarone as secondary prevention in patients with ICD was not only ineffective at reducing mortality but trended toward increasing SCD (RR 4.32; 95% CI 0.87 to 21.49) although with low quality evidence [52]. It is important to keep this apparent increase in mind during clinical decision making, especially in the context of amiodarone’s well known pulmonary, hepatic, and thyroid toxicities.

While amiodarone has the strongest evidence base for its routine use in clinical practice, it is often avoided due to the frequency of its long-term side effects. The high iodine content within its formulation can lead to thyroid toxicity, and both thyrotoxicosis and hypothyroidism can occur [91]. Amiodarone can also cause direct hepatotoxicity through phospholipase A inhibition and can cause a wide range of effects from liver function test abnormalities to fulminant liver failure [90,91]. Lastly, pulmonary fibrosis can also be a major complication in long-term amiodarone use and lead to significant morbidity and mortality [90]. Due to these significant side effects, patients on long-term use should be on routine surveillance of thyroid, hepatic, and pulmonary function.

### 5.2. Dronedarone

Dronedarone is non-iodinated benzofurane derivative structurally related to amiodarone [99]. Due to the similarity of its chemical structure, dronedarone has a similar mechanism of action to amiodarone through the blockade delayed rectifier potassium channel and increase in the ERP with subsequent decreased likelihood of reentry. However, the non-iodinated formulation subsequently has significantly lower rates of thyroid toxicity [100]. Initial studies have also found that dronedarone may have lower rates of pulmonary fibrosis, although hepatotoxicity rates are similar [100,101]. To date, most large trials comparing amiodarone to dronedarone for efficacy have focused on atrial fibrillation management. The largest evidence base for dronedarone use comes from a retrospective study by Friberg et al. who analyzed retrospective data on 45,000 patients who had AAD management for diagnosed AF. This analysis found that dronedarone had the lowest rates of sustained VA as well as cardiovascular death [102]. Further randomized trials are needed to evaluate the efficacy of dronedarone. Notably, dronedarone use in many patients with VA is limited due its increased mortality in advanced heart failure [103]. While dronedarone remains a viable alternative to amiodarone in relation to specific side effects, this mortality difference has led to its limited use in clinical practice in patients with abnormal heart structure.

### 5.3. Dofetilide

Dofetilide is a class III antiarrhythmic that selectively blocks the delayed outward rectifying potassium current, thereby increasing the effective refractory period (ERP) without delaying intracardiac conduction [104]. Importantly, dofetilide lacks the “out of class” effects of other class III antiarrhythmics and does not have significant hemodynamic consequences. In contrast to class I medications, dofetilide exhibits reverse use dependence, increasing effectiveness at lower heart rates [105]. Several trials including SAFIRE-D, EMERALD, and a subgroup analysis of the DIAMOND consistently demonstrate improved rates of cardioversion and maintenance of sinus rhythm in patients with atrial fibrillation and atrial flutter [104,105,106]. While the efficacy in management of atrial tachyarrhythmias is well established and approved for use by the FDA, the utility in ventricular tachycardia is less understood. Two large, randomized trials of dofetilide vs. placebo, DIAMOND-CHF and DIAMOND-MI, demonstrated no significant mortality difference and similar rates of ventricular tachycardia in both groups [107,108]. In a dose-ranging study by Echt et al., 8 of 18 patients with inducible VT and/or VF no longer had inducible ventricular arrhythmia after dofetilide infusion [109]. Similarly, Bashir et al. treated 50 patients with sustained monomorphic VT with a range of dofetilide doses, and 41% experienced suppression or slowing of inducible VT [110]. Dofetilide has been compared against sotalol in a double-blind crossover study of 135 patients with ischemic heart disease and inducible sustained VT; after oral loading for 3–5 days, 35.9% of patients responded to dofetilide where VT could not be induced [111]. At one year follow up of 41 patients, 7% had recurrence of VT and were therefore considered as treatment failure. Most recently, dofetilide demonstrated reduced episodes of VT and/or VF, as well as decreased ICD interventions in patients who had an ICD placed for secondary prevention and have failed other antiarrhythmics, including amiodarone [106]. The primary safety concern of dofetilide is QTc prolongation and torsades de pointes, and therefore, initiation should be monitored in the hospital setting. While not currently approved for ventricular arrhythmias, early evidence suggests that it is efficacious at suppressing inducible VT and decreasing clinically relevant ICD interventions.

### 5.4. Sotalol

Sotalol is a racemic mixture of d- and l-sotalol with unique pharmacologic effects exhibiting class II (nonselective ß-blocker) properties in addition to class III inhibition of delayed potassium rectifier channel, resulting in an increase in action potential duration and effective refractory period [112]. In contrast to amiodarone, and similarly to dofetilide, sotalol exhibits reverse use dependance and increases potency at lower heart rates. Class III antiarrhythmics became increasingly investigated for suppression of ventricular ectopy after the CAST study reported increased mortality of class I antiarrhythmics in patients with structural heart disease [109]. Sotalol demonstrated early efficacy in suppression of ventricular ectopy and suppression of inducible ventricular tachyarrhythmias [113,114,115]. The relative efficacy of sotalol was unknown until the ESVEM trial compared serial efficacy of seven antiarrhythmic medications through electrophysiologic testing and Holter monitoring in patients with ventricular tachyarrhythmias [19]. Sotalol was the most frequently efficacious mediation, in 43% of patients, with the lowest rate of arrhythmia recurrence and improvement in all-cause mortality. The mechanism for the mortality benefit reported in EVSEM was suspected to be related to beta-blocking effects of sotalol, this was further investigated in SWORD where patients with left ventricular dysfunction with recent or remote myocardial infarction were randomized to d-sotalol, which exhibits pure class III effects, or placebo [116]. SWORD was terminated early due to increased mortality in the d-sotalol group, with relative risk of 1.65, attributed to proarrhythmic effects of class III agents, most notably torsades de pointes. Shortly thereafter, results of the AVID trial comparing antiarrhythmic therapy, predominately amiodarone, to implantable cardioverter-defibrillator (ICD) revealed superiority favoring ICD placement. Many patients require antiarrhythmic therapy to decrease the frequency of ICD shocks—sotalol has demonstrated efficacy in reducing incidence and frequency of shocks in patients with an ICD for secondary prevention of VA71 [117]. In a randomized trial by Pacifico et al., the average number of annual shocks was significantly reduced in patients with sotalol therapy (1.43 vs. 3.89 shocks per year, *p* = 0.008) [117]. Due to this effect, sotalol remains a guideline-recommended therapy for secondary prevention in ICM patients who cannot tolerate amiodarone [1]. Although a current limitation is the requirement of frequent EKG monitoring in the inpatient setting for initiation of therapy to prevent torsades de pointe, the ongoing DASH-AF may negate this requirement. Lastly, in patients with arrhythmogenic right ventricular cardiomyopathy, sotalol has been shown to be superior in the suppression of inducible and non-inducible ventricular tachycardia [118].

## 6. Class IV Medications: Calcium Channel Blockade

The non-dihydropyridine calcium channel antagonists, verapamil and diltiazem, exhibit antiarrhythmic effects predominately at the AV-node via blocking of slow inward Ca current, thereby prolonging the effective refractory period (ERP) with minimal effects on atrial/ventricular myocytes or the His–Purkinje system [119]. With isolated effects at the AV-node, the efficacy of these agents at terminating and controlling ventricular rates in supraventricular tachycardias is not surprising, but the utility in ventricular tachycardia is less clear. Early studies by Wellens et al. and Belhassen and Horowitz demonstrated a lack of efficacy of verapamil in the termination of chronic recurrent ventricular tachycardia, as the arrhythmogenic mechanism is typically ventricular reentry circuit [120,121]. However, in 1981, Belhassen reported a unique VT with RBBB and left axis deviation that was repeatedly terminated and suppressed by verapamil, but not other antiarrhythmic agents, in a young adult without underlying structural heart disease [122]. This verapamil-sensitive form of VT is now defined as idiopathic left ventricular tachycardia (ILVT) with an electrophysiologic mechanism of macro-reentry circuit of normal Purkinje and abnormal Purkinje tissue [123,124]. Calcium channel antagonists have utility in another unique form of VT, CPVT, an inherited tachycardia in young individuals with structurally normal hearts at periods of increased sympathetic activity [125]. The cornerstone of therapy for CPVT remains ß-adrenergic blockers, however Rosso et al. demonstrated decreased exercise-induced ventricular ectopy and non-sustained VT with the addition of verapamil to beta blocker therapy and therefore serves as an adjunct in patients with persistent symptoms on beta blockers [126]. Overall, calcium channel antagonists have limited utility in the most common forms of VT but serve as a useful adjunctive therapy in CPVT and are first line agents for ILVT.

## 7. Medications Outside of the Classification System

### 7.1. Ranolazine

While most well known as a novel antianginal therapy, ranolazine exhibits features most similar to amiodarone, blocking inward depolarizing and outward repolarizing currents affecting sodium, potassium, and calcium channels. The net effect is a concentration-dependent prolongation of action potential duration and an early decrease after depolarizations [127]. Initially explored as a new agent in the control of atrial fibrillation, there is emerging evidence in its role in the prevention of VAs [128]. The largest trial to date, the RAID trial, showed a significant reduction in recurrent VAs in patients treated with ranolazine in high-risk patients (hazard ratio 0.70, *p* = 0.03) [129]. The MERLIN-TIMI 36 trial showed that when ranolazine was added to post-NSTEMI care, patients saw a 40% relative risk reduction in sustained VA on 7-day continuous EKG monitoring [130]. Furthermore, a study of 12 patients with refractory VAs showed that when patients had ranolazine added to their existing AAD regimen, 93% were VA-free at 6 month follow up [131].

Given its effect on inward sodium current and the subsequent increase in ventricular relaxation and perfusion, ranolazine has emerged as a therapeutic option in the treatment of HCM. The RESTYLE-HCM study showed that there was greater than 50% reduction in PVC burden with treatment versus placebo over five months [132]. While this study measured improvement in the burden or recurrence of VAs in this population, there is no study that directly investigates this concern. The reduction of PVC burden may act as a proxy for VA risk and is a focus for further investigation.

### 7.2. Adenosine

Adenosine is an endogenous nucleoside that acts on adenosine receptors primarily located on the specialized conduction tissues of the SA and AV nodes, resulting in activation of potassium channels and hyperpolarization and decreased automaticity of these tissues. With a half-life of only 10 s, this classically results in a transient AV nodal block to terminate SVT. However, in some forms of repetitive monomorphic ventricular tachycardia (RMVT), specifically in right ventricular outflow tract (RVOT) VT, adenosine is such an effective treatment that these arrhythmias are also known by the name “Adenosine-Sensitive VT” [133,134,135]. While these patients often have outpatient treatment with calcium channel blockers, adenosine is the gold standard treatment in those requiring inpatient treatment or with life-threatening arrhythmias [134]. In a 2014 study by Lerman et al., patients who underwent adenosine treatment for VT in the setting of structural disease had a 0% response rate, while those who underwent adenosine treatment for VT with known focal RVMT had a 95% success rate in VT termination [136]. This study highlights that adenosine can be a highly successful tool when used only in the correct setting.

### 7.3. Digoxin

Digoxin exhibits both mechanical and electrophysiologic effects on the cardiovascular system: the former through inhibition of Na^+^/K^+^ ATPase with a resultant rise in intracellular calcium and increased contractility and the latter through enhancing vagal tone predominately at the AV node, resulting in decreased conduction velocity and increased ERP. Digoxin use has been largely relegated to advanced heart failure and refractory atrial fibrillation, as initial trials showed a trend toward increased mortality in these populations [137]. Further meta-analyses have re-demonstrated these results [138,139]. Although digoxin has been shown to be a risk factor in the development of VA, there were initially no direct studies to quantify this risk [140]. In 2015, a sub-set analysis of the MADIT-CRT by Lee et al. showed that digoxin had a 41% increased risk of VT/VF as compared to other patients [141]. Patients with advanced heart failure or prior VA history should be cautioned when considering digoxin therapy.

### 7.4. Isoproterenol

Isoproterenol is a beta-1 and beta-2 adrenergic receptor agonist that activates cardiac pacemaker cells [142]. Given the pro-sympathetic effects of the medication as a beta agonist, it is avoided in typical VA without an underlying channelopathy. However, isoproterenol can be an effective agent in the treatment of electrical storm in Brugada Syndrome patients through its influence on I_CA-L_ channels. Given the small patient population of this diagnosis, studies are largely case series but do show strong efficacy in the use of Isoproterenol [143,144].

### 7.5. Ivabradine

Ivabradine functions in a use-dependent fashion at the SA node, inhibiting the mixed sodium-potassium current (If) and thereby slowing depolarization of the pacemaker potential and lowering heart rate without affecting inotropy or vascular resistance [145]. Initially investigated as an anti-anginal in the BEAUTIFUL trial, and subsequentially in the SIGNIFY trial, ivabradine did not improve cardiovascular outcomes in patients with stable coronary artery disease [146,147]. Ivabradine is approved as an adjunctive medication to GDMT in patients with HFrEF and NYHA class II-III symptoms following the results of the SHIFT trial, which demonstrated a reduction in heart failure hospitalizations and subsequently cardiovascular death [148]. Given its unique electrophysiologic effects to reduce heart rate, a small randomized crossover trial of 21 patients found improvement in symptoms if patients with inappropriate sinus tachycardia [149]. Recent case reports document ivabradine as an effective adjunctive agent in CPVT refractory to nadolol, flecainide, and cervical sympathectomy [150,151]. More research is needed before widespread implementation in CPVT, but ivabradine should be considered in refractory cases. Most adverse effects of ivabradine are related to symptomatic bradycardia; however, a meta-analysis demonstrated a relative risk of 1.15 for the development of atrial fibrillation, and associations of torsades de pointes with concomitant use of QT-prolonging agents has been described in case reports [152].

## 8. Conclusions

Amiodarone remains the most evidence-based treatment of recurrent VAs. Beta blocker therapy is likely undervalued given the strength of evidence behind it and should be considered a cornerstone of VA pharmacologic therapy. There is limited but relatively strong data to support the use of specific antiarrhythmics in specialized circumstances (i.e., quinidine in Brugada, mexiletine in Long QT, beta blockers in CPVT and Long QT, etc). The medication choices in each of these specialized circumstances is listed in Table 2 above with corresponding evidence. Anti-arrhythmic drugs are no replacement for ICD, which have been consistently shown to be the only therapy to consistently reduce mortality from VAs. However, there is a large body of evidentiary support that AADs should be considered in reducing the morbidity of ICD therapies.

## Figures and Tables

**Figure 1 jcm-11-03233-f001:**
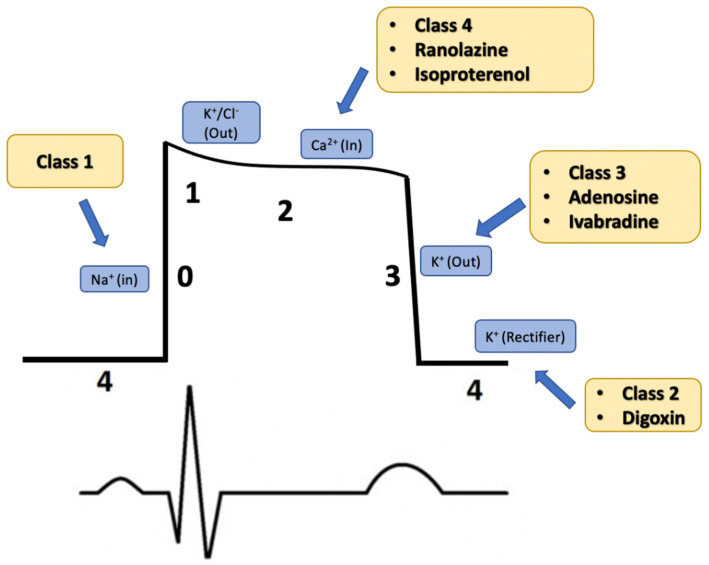
Anti-arrhythmic medications grouped by class and illustrated by the effect the cardiac action potential. Each number reflects a change in ion channels: “0” responding to influx of sodium, “1” and “3” corresponding to repolarizing efflux of potassium, “2” corresponding influx to calcium, and “4” corresponding to baseline potential through the Na/K pump.

**Table 1 jcm-11-03233-t001:** Pharmacologic characteristics of all medications by Vaughan–Williams classification, including major side effects and contraindications.

Drug	Mechanism of Action	Channels Effected	Dosing	Contraindications	Important Side Effects/Considerations
Class I—Sodium Channel Agents					
**Class IA**					
Quinidine	• Blockade of the rapid inward sodium depolarization current in a use dependent fashion and prolongation of repolarization via blockade of the delayed rectifier potassium channel in reverse use-dependence fashion.	INa, Ito, IKr, M, α	PO only: Sulfate 300 mg-max tolerated q6h–q12h, Gluconate 324–648 mg q8h–q12h	Severe AV node dysfunction, Thrombocytopenia or underlying platelet dysfunction. Prolonged QT.	• Negative Ionotropy, profound hypotension.
	• They slow phase 4 depolarization during spontaneous automaticity.				• Can be pro-arrhythmogenic at therapeutic doses
	• The net effect is preferential prolongation of action potential duration at fast heart rates, prolonged effective refractory period (ERP), and decreased automaticity				• Can prolong QT
Procainamide		INa, IKr	IV: 10–17 mg/kg; PO: 500–1000 mg q6h	Severe AV node dysfunction, underlying Systemic Lupus Erythematosus. Prolonged QT.	• Can cause Drug-induced Systemic Lupus Erythematosus.
					• Can cause agranulocytosis at therapeutic doses requrining CBC monitoring
					• Breaks down into toxic metabolite ‘NAPA’ which requires monitoring, especially when on IV formulations.
					• Can cause negative inotropy and profound hypotension.
					• Can be pro-arrhythmogenic at therapeutic doses.
					• Can prolong QT
Disopyramide		INa, Ito, IKr, IK(ATP), M	PO only: 150 mg q6h	Severe AV node dysfunction	• Negative Ionotropy, profound hypotension.
					• Can be pro-arrhythmogenic at therapeutic doses
					• Significant Anticholinergic side effects
					• Can prolong QT
**Class IB**					
Lidocaine	• Use dependent blockade of the inward sodium depolarization current thereby decreasing maximal velocity of depolarization.	INa	IV only: 1 mg/kg bolus followed by 1–3 mg/min	Severe AV node dysfunction	• CNS side effects including seizures, coma, or death requiring frequent blood level monitoring.
Mexiletine	• Shortening of action potential and ERP duration thereby decreasing automaticity of phase 4 depolarization.	INa	PO only: 150 mg q8h		• Higher incidence of Drug-Induced Liver Injury.
					• Can cause tremors and ataxia
					• Has high rates of GI distress.
**Class IC**					
Flecainide	• Most potent among the inward sodium blocking agents thereby markedly reducing the action potential conduction velocity in atrial, ventricular, and specialized conduction tissues.	INa, IKr, IKur	PO only: 50–200 mg q12h (can increase to q8h)	Structural Heart Disease or Reduced Ejection Fraction	• Can be pro-arrhythmogenic at therapeutic doses.
	• Blocking occurs in a use-dependent fashion with minimal effect on overall action potential duration or ERP.				• PR and QRS prolongation
Propafenone		INa, IKr, IKur, β, α	PO only: IR release 150–300 mg q8h; ER release: 225–425 mg q12h		• May cause the slowing of atrial arrhythmias leading to dangerous 1:1 conduction.
**Class II—Beta Blockers**					
Propranolol	• Blunting sympathetic activity on cardiac tissue, most notably through decreasing phase 4 depolarization thereby decreasing automaticity via decreased conduction velocity and increased ERP within the AV-node decreasing reentry	β1, β2, INa	IV: 1–3 mg boluses q5min, PO: 10–160 mg q6h–q12h	Severe AV node dysfunction, Sick Sinus Syndrome	• Can cause severe bradycardia and precipitate cardiogenic shock
Metoprolol		β1	IV: 5mg q5min ×3, PO: Tartrate 12.5–200 mg q6h–q12h, Succinate 12.5–200 mg q12h–q24h		
Nadolol		β1, β2	PO only: 40–320 mg qDay		
Carvedilol		β1, β2, α	PO only: 3.125–25 mg q12h		
**Class III—Potassium Channel Agents**					
Amiodarone	• Primarily through blockade of the delayed rectifier potassium channel effectively prolonging repolarization and increasing the ERP to decrease the likelihood of reentry.	INa, ICa, IKr, IK1, IKs, Ito, β, α	IV: 150–300 mg bolus, 0.5–1 mg/min (1 mg/min for 6 h then 0.5 mg/min for 18 h), PO: initial 400 mg q12h then taper to as low as 100 mg q24h if needed	Pre-existing Thyroid, Liver, and Pulmonary Disease	• Requires biannual TSH and LFT monitoring for thyroid/liver toxicity.
	• Additionally, amiodarone has mechanistic overlap with all other classes of antiarrhythmics and has vasodilatory and negative inotropic effects.				• Requires annual PFT monitoring for pulmonary fibrosis.
					• Can cause skin photosensitivity.
					• Can cause corneal microdeposits effecting vision
Sotalol	• Sotalol is a racemic mixture of d- and l- sotalol with unique pharmacologic effects exhibiting class II (nonselective ß-blocker) properties.	IKr, β1, β2	IV: 75 mg q12h, PO: 80–120 mg q12h	Prolonged QT	• Profound QT prolongation, must complete load under observation in hospital with EKG monitoring (although emerging data to support rapid IV loading)
	• Additionally, to class III inhibition of delayed potassium rectifier channel resulting in an increase in action potential duration and effective refractory period.				• In patients with advanced heart failure, Sotalol can precipitate cardiogenic shock.
Dofetilide	• Specific class III antiarrhythmic which blocks the delayed outward rectifying potassium current thereby increasing the effective refractory period (ERP) in a reverse use-dependence fashion without delaying intracardiac conduction.	IKr	PO only: 500 mcg q12h		
**Class IV—Calcium Channel Blockers**					
Verapamil	• The non-dihydropyridine calcium channel antagonists, verapamil and diltiazem, exhibit antiarrhythmic effects predominately at the AV-node via blocking of slow inward Ca current.	ICa-L	IV: 2.5–5 mg q15–30 mins as tolerated, PO: IR release 120 mg q8h; ER release: 120–480 mg q12h–q24h	Severely Depressed EF (<35%), Severe AV node dysfunction	• Negative inotropy, can precipitate cardiogenic shock.
	• Blocking of inward Ca current thereby prolongs the effective refractory period (ERP) with minimal effects on atrial/ventricular myocytes or the his-purkinje system				
Diltiazem	• Although less common, these agents can cause blockade of slow inward calcium channels on some sensitive fascicular tissues.	ICa-L	IV: 0.25 mg/kg bolus followed by 5–15 mg/h as tolerated, PO: IR release: 30–120 mg q6h–q12h, ER release: 30–240 mg q12h–q24h	Severely Depressed EF (<35%), Severe AV node dysfunction	• Negative inotropy, can precipitate cardiogenic shock.
**No Class in Vaughn-Williams**					
Ranolazine	• Ranolazine exhibits features most similar to amiodarone, blocking inward depolarizing and outward repolarizing currents affecting sodium, potassium, and calcium channels.	INa, IKr	PO only: 500–1000 mg q12h	Hepatic Cirrhosis	• Can prolong QT
	• The net effect is a concentration dependent prolongation of action potential duration and decreased in early after depolarizations.				
Ivabradine	• Ivabradine functions in a use-dependent fashion at the SA node inhibiting the mixed sodium-potassium current (If) thereby slowing depolarization of the pacemaker potential and lowering the heart rate	If	PO: 2.5–5 mg q12h	Bradycardia, heart block, sick sinus syndrome	• Symptomatic bradycardia, increase risk of atrial fibrillation
Adenosine	• Adenosine is an endogenous nucleoside that acts on adenosine receptors primarily located on the specialized conduction tissues of the SA and AV nodes resulting in activation of potassium channels and hyperpolarization and decreased automaticity of these tissues.	Activation of A1, A2, and IKATP	IV only: initial 6mg dose followed by 12 mg × 2 q1 minutes if peripherally administered.	Use in pre-excitation syndromes (Wolff-Parkinson-White) can precipitate VA’s.	• Can cause temporary, but profound, chest discomfort.
			50% dose reduction if via central line		

**Table 2 jcm-11-03233-t002:** Pharmacologic therapy in ventricular arrhythmia syndromes with structurally normal hearts.

Arrhythmia Type	Etiology	Medication	Evidence
**Brugada Syndrome**	Brugada syndrome is an autosomal dominant genetic disorder with variable expression characterized by abnormal findings on the surface electrocardiogram (ECG) with an increased risk of ventricular tachyarrhythmias and sudden cardiac death. Most commonly, Brugada is the result of defective sodium channels leading the reduction of sodium inflow current and a subsequent reduction in the duration of action potentials.	First Line: Quinidine	Belhassen et al. [32]
			Marquez et al. [31]
**Long QT Syndrome**	Long QT syndromes may be congenital or acquired and represent a disorder of myocardial repolarization characterized by a prolonged QT interval on the electrocardiogram (ECG). These findings lead to an increased risk of polymorphic VT, which can be life-threatening. This review focuses on congenital LQT in the structural normal heart.		
Long QT1/Long QT2	Both Long QT1 and Long QT2 are caused by mutations in genes encoding potassium channel leading to a defect in inward potassium current and QT prolongation. Their major difference is LQT1 effects KCNQ1 gene most commonly leading to a defect in slow potassium current (Iks) while LQT2 effects KCNH2 gene most commonly leading to a defect in rapid potassium current (Ikr). The overall effect and treatment remains the same for both types.	Beta Blockers (Propranolol/Nadolol preferred)	Bennett et al. [88]
			Schwartz et al. [87]
Long QT3	Unlike LQT1/LQT2, LQT3 is caused by a defect in SCN5A gene most commonly. This mutation leads to a defect in a cardiac sodium channel and subsequently increases the delayed Na+ inward current and, therefore, prolonging the action potential duration. As the primary effect is on sodium channels, it is amenable to treatments with mechanism of action on these channels.	First Line: Beta Blockers	Wilde et al. [89]
		Second Line: Mexiletine	Mazzanti et al. [56]
		Salvage Therapy: Flecainide/Propafenone	Moss et al. [63]
			Belardinelli et al. [62]
**Catecholaminergic Polymorphic Ventricular Tachycardia (CPVT)**	CVPT is an often familial syndrome leading to exercise-induced polymorphic VT in childhood/adolescents. CVPT most commonly arises due to mutations in one of two genes: the cardiac ryanodine receptor gene (an autosomal dominant form) and the calsequestrin 2 gene (autosomal recessive). Both mutations act by inducing intracellular calcium release and causing a intracellular calcium overload. This overload leads to delayed afterdepolarization, which can induce ventricular arrhythmias.	First Line Beta Blockers (Nadolol with strongest evidence)	Priori et al. [86]
		Second Line: Calcium Channel Blockers	Leenhardt et al. [85]
			Rosso et al. [126]
**Idiopathic Left Ventricular Tachycardia (ILVT)**	ILVT is an idiopathic form a VT presenting in young adulthood of unclear etiology. Current studies indicate that this arrhythmia is caused by localized reentry circuit close to the posterior fascicle.	Calcium Channel Blockers	Belhassen et al. [121]
			Ohe et al. [123]

## Data Availability

Not applicable.

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
