# Peer review of "Pharmacologic Management for Ventricular Arrhythmias: Overview of Anti-Arrhythmic Drugs"

_jcm, 2022, doi:10.3390/jcm11113233_

Round 1
Reviewer 1 Report
Larson et al has written a review article in regard to use of AADs in VT settings. Several parts should be amended.
- page 2 paragraph 76-78, can the authors describe more for the term "VT termination"? It was mentioned that 67% in Procainamide vs 38% in Amiodarone but the time to termination was not described. Without clearly stating, this poses a potential lead time bias for readers.
- page 3 paragraph 111, the authors described Brugada syndrome as a specific VA condition which is triggered by ectopy. Is this statement well validated? Please cite the relevant articles if any. Main pathophysiology of VA development in Brugada syndrome is currently proposed by dispersion and inequalities in ventricular depolarization and repolarization, leading to reentry.(https://doi.org/10.1016/j.cardiores.2005.03.005) Otherwise, it remains poorly understood.
- page 4 paragraph 145, can authors elaborate more information in regard to use of lidocaine in MI patients as this is very common in clinical practice?
- Page 6 paragraph 252, there is a statement narrating that nonselective bb has superior outcomes to selective bb. What are the outcomes authors tried to point out, VA, mortality? If VA, in what settings do patients merit nonselective BB prior to selective BB?
- Page 6 under amiodarone section, authors should add that amiodarone is not reverse use dependent like others in class III
- Under the amiodarone section, authors should state and cite any references that despite its prolonging QT interval, incidence of VA is not as common as other AADs
- Under sotalol section, authors should state that this medication is reverse use dependent
- Under class III AADs discussion, I think Drodenarone is missing.
- As no tables are attached into this file, table summarizing medications described in this article, doses and their side effects should be provided unless it is already there
Author Response
1. page 2 paragraph 76-78, can the authors describe more for the term "VT termination"? It was mentioned that 67% in Procainamide vs 38% in Amiodarone but the time to termination was not described. Without clearly stating, this poses a potential lead time bias for readers.
-
- Definition from study was added to the review.
2. page 3 paragraph 111, the authors described Brugada syndrome as a specific VA condition which is triggered by ectopy. Is this statement well validated? Please cite the relevant articles if any. Main pathophysiology of VA development in Brugada syndrome is currently proposed by dispersion and inequalities in ventricular depolarization and repolarization, leading to reentry.(https://doi.org/10.1016/j.cardiores.2005.03.005) Otherwise, it remains poorly understood.
-
- The prior wording was misleading and confusing, the manuscript is trying to illustrate that ectopy can have a role in triggering sustained VA’s in Brugada patients, not that this is the underlying pathophysiology. Current edit attempts to clarify this point.
3. page 4 paragraph 145, can authors elaborate more information in regard to use of lidocaine in MI patients as this is very common in clinical practice?
-
- Meta-analysis discussion now included in this paragraph.
4. Page 6 paragraph 252, there is a statement narrating that nonselective bb has superior outcomes to selective bb. What are the outcomes authors tried to point out, VA, mortality? If VA, in what settings do patients merit nonselective BB prior to selective BB?
-
- Outcomes included from trial: hospital days, ICU stay, and incidence of VA. There was no difference in mortality. Directly supported the use of non-selective BB in electrical storm and CVPT as noted now in the review.
5. Page 6 under amiodarone section, authors should add that amiodarone is not reverse use dependent like others in class III
-
- This description is now included in the review.
6. Under the amiodarone section, authors should state and cite any references that despite its prolonging QT interval, incidence of VA is not as common as other AADs
-
- This effect is now noted and cited with individual studies within the review.
7. Under sotalol section, authors should state that this medication is reverse use dependent
-
- This is now included in the sotalol section
8. Under class III AADs discussion, I think Drodenarone is missing.
-
- There is now a section on drodenarone.
9. As no tables are attached into this file, table summarizing medications described in this article, doses and their side effects should be provided unless it is already there
-
- Apologies as there was an issue in submission, the tables so now be included with the revision.
Reviewer 2 Report
Dear Editor dear Authors, thank you for the opportunity to review the article „Pharmacologic Management for Ventricular Arrhythmias: Overview of Anti-Arrhythmic Drugs.“ In general it is a interesting review article.
I heve following suggestions:
- Consider that one of the most "popular" antiarrhythmic (AA) drugs are beta-blockers, please discuss which beta-blocker is better to prescribe for various arrhythmias? what data do we currently have?
- When describing amiodarone, please describe also the side effects in detail! This is something that happens often and it will be important for our young colleagues.
- Please describe the role of isoproterenol in the treatment of electrical storms in Brugada syndrome.
- To make the review more spectacular, please insert a picture of the action potential change for each AA group.
Best regards
Author Response
1. Consider that one of the most "popular" antiarrhythmic (AA) drugs are beta-blockers, please discuss which beta-blocker is better to prescribe for various arrhythmias? what data do we currently have?
-
- There are now specific discussions regarding non-selective beta blockers and their evidentiary support in treatment of specific VA’s.
2. When describing amiodarone, please describe also the side effects in detail! This is something that happens often and it will be important for our young colleagues.
- These is now a detailed paragraph on Amiodarone side effects.
3. Please describe the role of isoproterenol in the treatment of electrical storms in Brugada syndrome.
- A separate section was added with data supporting the use of isoproterenol in Brugada patients.
4. To make the review more spectacular, please insert a picture of the action potential change for each AA group.
Image now included below table 1.
Round 2
Reviewer 1 Report
Authors have clarified all my queries properly.
Reviewer 2 Report
Congratulations to the authors!